# Uninterrupted Classroom Sitting is Associated with Increased Discomfort and Sleepiness Among College Students

**DOI:** 10.3390/ijerph16142498

**Published:** 2019-07-13

**Authors:** Katie R. Hosteng, Alison Phillips Reichter, Jacob E. Simmering, Lucas J. Carr

**Affiliations:** 1Department of Health and Human Physiology, University of Iowa, Iowa City, IA 52242, USA; 2Pulmonary, Critical Care, and Occupational Medicine, Department of Internal Medicine, University of Iowa, Iowa City, IA 52242, USA

**Keywords:** sedentary behavior, discomfort, sleepiness, college students

## Abstract

Acute bouts of uninterrupted sitting has been associated with discomfort and fatigue in adult populations. However, little is known regarding the impact of uninterrupted sitting on such outcomes among college students. Understanding these relations would be useful for informing best practice and future interventions. The present study explored the relation between uninterrupted sitting and perceived levels of physical discomfort and sleepiness among college students in a real classroom setting. We recruited 54 undergraduate students enrolled in a single class at a Midwestern university. Participants remained seated throughout a 2.5 h lecture while completing the Stanford Sleepiness Scale (SSS) and General Comfort Scale (GCS) every 15 min. Linear mixed effect model analyses were used to determine the relations between the independent and dependent variables and the duration at which students reported significant impairments in discomfort and/or sleepiness. Classroom sitting time was associated with increases in discomfort (*r* = 0.28, *p* < 0.01) and sleepiness (*r* = 0.30, *p* < 0.01). Students reported significant impairments in discomfort and sleepiness after 75 and 15 min, respectively. These findings support further research into the acceptability, feasibility and efficacy of interventions designed to interrupt classroom sitting on discomfort, sleepiness and measures of academic performance.

## 1. Introduction

The 2018 Physical Activity Guidelines for Americans reported on the health consequences of sedentary behavior for the first time. The report concluded that there is strong evidence that demonstrates long-term exposure to high amounts of sedentary behavior (time spent sitting or reclining that requires little energy expenditure), significantly increases the risk of all-cause mortality, cardiovascular disease incidence and mortality, and type 2 diabetes incidence [1]. The guidelines also recommend that individuals who fail to meet the physical activity guidelines should replace sedentary behavior with light-intensity physical activity to reduce the risk of all-cause mortality. While it is significant that the Physical Activity Guidelines mentioned sedentary behavior in the report for the first time, the guidelines in this area are still vague due to the limited amount of evidence available on the dose-response relationship between sedentary behavior and many health outcomes. Additional research in needed in this area in order to inform guidelines on sedentary behavior and interventions targeting sedentary behavior. 

Beyond the long-term consequences of chronic sedentary behavior, recent studies exploring the possible benefits of interrupting acute bouts of uninterrupted sitting have reported several health-related consequences to acute bouts of uninterrupted sedentary behaviors. These studies have mostly been conducted in controlled laboratory settings, have focused on cardiometabolic risk factors, and have included middle-aged, sedentary adults. For example, Wenneberg et al. (2016) reported significant increases in reported levels of fatigue and impaired cognitive function after four hours of continuous sitting [2]. Similarly, in a study of 23 middle-aged, overweight/obese office workers, Thorp and colleagues found those who worked in a mostly seated position for eight hours reported significantly higher levels of fatigue and musculoskeletal discomfort compared to a group that worked the same duration but interrupted their sitting every 30 min with a brief bout of standing [3]. Our team recently demonstrated sitting for four continuous hours significantly increased reports of physical discomfort and sleepiness among a sample of 15 middle-aged, sedentary office workers [4]. While these findings suggest uninterrupted bouts of sitting may negatively impact self-reported levels of discomfort and mental alertness, little research in this area has been conducted among other populations and/or in real-world settings. 

College students are a large, yet understudied population that may be at increased risk for sedentary-related consequences [5]. According to the National Center for Education Statistics, 19.9 million students were enrolled in colleges/university in the U.S. in 2018 [6]. Data from the 2015 National College Health Assessment suggest college students sit as much as 30 h per week [7]. A study by Johnston and colleagues found college student’s daily sitting time significantly increased from their freshmen year (329.6±192 min/day) to their senior year (405.2±240.3 min/day) [8] which supports the theory that the college years are a particularly important time to intervene on sedentary behaviors [9]. A large portion of college student’s sitting time is accumulated while in class, making the classroom an ideal environment for observational and intervention research [10]. Our previous work has shown that most college students (82.7%) report sitting for the entire duration of their class time, which is primarily due to a lack of opportunities to stand and move during class. Additionally, most students (95%) reported they would prefer the option to stand during class [11] and more than half of all students felt having the opportunity stand during class would positively impact that health, attention, and feelings of restlessness [11]. Unfortunately, no studies to our knowledge have examined the relationship between classroom sitting and any sedentary related outcomes among college students. 

Physical comfort/discomfort and mental alertness are two outcomes that have previously been associated with sedentary behavior and may influence college student’s abilities to perform in class. Discomfort is defined as an unpleasant, acute state of the human body in reaction to its physical environment (p. 271) [12]. Discomfort can also be experienced as a chronic condition which is generally referred to as chronic pain. Alertness is defined as the ability to remain focused, able to concentrate, motivated, and aware of the environment [13]. Alertness is the opposite of sleepiness and is considered to be a short term state. 

It is currently not well understood how or if prolonged classroom sitting impacts either comfort/discomfort or alertness/sleepiness among college students. It is also not known how much uninterrupted classroom sitting is too much for college students. Answers to these questions could inform future guidelines, teaching practices, and interventions targeted to classroom sitting time. Therefore, the primary purpose of this study was to explore the relations between prolonged classroom sitting and students’ perceived levels of physical comfort/discomfort and alertness/sleepiness. We hypothesized longer bouts of classroom sitting would be positively associated with higher levels of perceived physical discomfort and sleepiness. A secondary aim was to determine the duration (minutes) of classroom sitting time necessary to induce significant impairments in perceived physical discomfort and sleepiness. Answering this second question will help inform future recommendations related to classroom sitting. Finally, as an exploratory aim, we sought to determine whether daily sitting time and/or self-reported pain moderated the relationship between classroom sitting time and comfort/alertness. We hypothesized students who sat more during the day prior to class and students who reported higher levels of physical pain would report higher levels of discomfort and sleepiness during class. 

## 2. Materials and Methods

### 2.1. Participants

A convenience sample of 54 undergraduate college students enrolled in a single 2.5-h lecture course (3:30 pm–6:00 pm) at a large Midwestern university in the U.S were recruited to participate. All participants were at least 18 years of age. After receiving approval from the course instructor, a research team member explained the study to all potential participants (e.g., students enrolled in the class) on the first day of class during the fall 2016 semester. After all questions related to participation in the study were addressed, interested participants were asked to sign an informed consent document. All subjects gave their informed consent for inclusion before they participated in the study. The study was conducted in accordance with the Declaration of Helsinki, and the protocol was approved by Human Subjects Office of the University of Iowa (project identification code 201611795). Students received no compensation for participation. Data collection was performed during class the following week.

### 2.2. Procedure

At the beginning of class on the day of data collection, participants completed a general demographic survey that asked about their age, weight (lbs.), height (feet and inches), sex, race, and ethnicity. Participants also reported the time they got out of bed that day (HH:MM), the total amount of time (hours: minutes) they had spent sitting that day in various domains (i.e., traveling, at home, school, work, or during leisure time), and whether they had experienced any pain or discomfort that had lasted at least two days over the previous 12 months in the neck, shoulders, elbow, wrist/forearm, hand, upper back, and/or lower back (yes/no). Pain experiencers were defined as individuals who reported pain in any area of the body over the past 12 months. Participants were also grouped into low (report sitting less than median sitting time) and high (reported sitting more than median sitting time) sedentary groups. 

On the testing day, participants were asked to remain seated throughout a 150-min lecture-style class with the exception of using the bathroom. The chairs students sat in during class were traditional seated classroom desks made of polyethylene (no cushion) with a fiberboard arm desk top. Classroom sitting time was measured via direct observation by a research staff member who observed the class. If a student stood up for any reason during class, the time (minute of class period) was documented for that individual. 

Every 15 min (eight times total), participants were prompted to report their momentary levels of physical comfort/discomfort and alertness/sleepiness. Momentary physical discomfort was measured using the 11-item General Comfort Scale (GCS) which has been demonstrated as a valid measure of discomfort while sitting [14]. GCS scores range from 0 = “I feel completely relaxed” to 10 = “I feel unbearable pain”. We determined a value of 4.0 as the critical threshold for significantly impaired comfort because participants transitioned from reporting a 3.0 (I feel barely comfortable) to 4.0 (I feel uncomfortable) on the GCS. Momentary alertness was assessed using the Stanford Sleepiness Scale (SSS) [15] which has been demonstrated as a valid measurement of momentary alertness among male college participants (r = 0.68) [16]. The SSS has also demonstrated a reliability of 0.88 [17,18]. The SSS uses an 8-point scale with scores ranging from 1= “Feeling active, vital, alert, wide awake” to 8 = “Asleep”. Consistent with the survey instructions, a value of 3.0 (awake, but relaxed; responsive but not fully alert) was considered the threshold for significantly impaired alertness.

### 2.3. Statistical Analysis 

Demographics (age, sex, body mass index, race, and ethnicity), waking time, daily sitting time, and pain data were analyzed using descriptive statistics and are summarized in Table 1. To determine the general relationships between classroom sitting time (minutes) and both perceived alertness and physical discomfort, we used Pearson product-moment correlations. Additionally, we modeled the change in comfort and alertness using a linear mixed effect model (LMM) that regressed the discomfort and sleepiness scores on the time since the start of class. The LMM included a random intercept by student to account to differences between students. 

In order to determine how long it took for participants to report significant impairments in discomfort and alertness, we used the Kaplan-Meier estimator to model the time until the subjects reached critical values for discomfort (4.0 = “I feel uncomfortable”) and sleepiness (3.0 = “awake, but relaxed; responsive but not fully alert”). The time until this occurs is a time-to-event measure so we used survival analysis. Students who stood up during the lecture were censored at the last reading they reported before standing. 

Finally, to test our exploratory aim of whether self-reported pain and/or daily sitting time prior to class influenced discomfort and/or alertness, we compared baseline and average levels of discomfort and sleepiness for pain/non-pain reports and low/high sedentary groups using a linear mixed effects model with bootstrapped confidence intervals. To determine whether self-reported pain and/or daily sitting time influenced the time to significant discomfort and sleepiness, we compared the differences in the survival curves between pain/non-pain reporters and low/high sedentary groups using Cox proportional hazards regression. 

All analyses were performed in R (version 3.4.1) [19]. The LMM were estimated using the lme4 (version 1.1-13) [20] package and the survival analysis was performed using survival (version 2.41-3) package [21].

## 3. Results

A total of 54 participants completed the study. Participants were mostly female, White and Non-Hispanic (Table 1). The observed class began at 3:30 pm. Participants reported sitting nearly three-quarters (73.9%) of their total wakeful time prior to class. Over two-thirds of participants (68.5%) reported experiencing significant pain in at least one area that lasted for at least two days over the past 12 months (Table 1).

An overall positive association was observed between classroom sitting time and discomfort, *r* = 0.28, *p* < 0.01 (see Figure 1 and Figure 2). Our comfort model confirms all students became less comfortable over the duration of the 2.5-h class (+0.84 comfort units / hour, 95% CI [0.59, 1.11]). Of the 54 students, 35 (64.8%) reported becoming uncomfortable (GCS value of 4.0) sometime before the class ended. Based on a Kaplan-Meier analysis, the median time until significant discomfort was 75.0 min (95% CI: [60, 90]). 

An overall positive association was also observed between classroom sitting time and sleepiness, *r* = 0.30, *p* < 0.01 (see Figure 3 and Figure 4). Our sleepiness model confirms all students became increasingly more sleepy (less alert) over the duration of the 2.5 h class (+0.63 attention units per hour, 95% CI [0.44, 0.87]). Of the 54 students, nearly all (n = 53, 98.1%) reported significant sleepiness (SSS value of 3.0) before the class ended. The median time until students reported significant sleepiness was 15 min (95% CI: [0, 30]). 

Students who reported pain (N = 37) also reported significantly higher levels of discomfort (2.8 vs. 1.8, *p* = 0.009) than non-pain reporters. Discomfort differences were maintained during the course with the average reported discomfort of 3.8 among those who experienced pain versus 2.5 among those who did not (*p* < 0.001). However, the rate at which the discomfort increased did not differ between the two groups (additional discomfort per hour among pain experiencers versus non-pain experiences was +0.33 with 95% CI −0.21 to +0.87). Pain did not significantly influence student’s reported levels of sleepiness during class. Students who reported pain did not enter the class in an any more sleepy state than their non-pain reporting peers (+0.33 attention units, 95% CI [−0.37, 1.01]) and did not increase their sleepiness more rapidly than their peers (+0.02 attention units per hour compared to non-pain students, −0.45 – 0.51). 

Time spent sitting prior to class did not significantly impact levels of discomfort during class. Students who sat more than the median prior to class (> 5 h) did not report higher levels of discomfort (−0.16, 95% CI [−0.94, 0.63]) or increase their discomfort at a faster rate as the class went on (difference in slopes = −0.13 per 6 min, 95% CI [−0.63, 0.38]) when compared to those who sat less than the median prior to class (<5 h).

Time spent sitting prior to class also did not affect any student’s reported levels of sleepiness during class. There was no difference between students who sat more versus less than the median in terms of time-until-inattention (hazard ratio = 1.62, 95% CI [0.93, 2.82]). Additionally, when controlling for the fact that those who sat more than the median amount of time were generally awake longer, the differences in survival become even smaller (hazard ratio for sitting more than median when time awake is included in the model is 1.33, 95% CI [0.75, 2.37]). 

## 4. Discussion

The findings of this study suggest prolonged and uninterrupted sitting during a traditional 2.5-h college lecture is associated with increased levels of self-reported physical discomfort and sleepiness among college students. Significant levels of physical discomfort were reported after 75 min of uninterrupted sitting while significant levels of sleepiness were reported after just 15 min. Additionally, students who reported experiencing significant pain in the past year reported higher levels of discomfort entering class and reported significant discomfort after just 60 min of uninterrupted class time sitting. Neither pain nor high sitting time accumulated during the day prior to class had a significant effect on reported levels of alertness. 

These findings advance our understanding of the possible impact uninterrupted classroom sitting time has on outcomes related to student performance. No studies, to our knowledge, have observed the effects of acute bouts of uninterrupted classroom sitting on musculoskeletal discomfort and/or alertness among college students. Still, our findings are consistent with our previous study, which found uninterrupted sitting was associated with significant increases in discomfort among 15 middle-aged sedentary office workers in a laboratory setting [4]. While both studies used the same measure of discomfort, sedentary office workers did not report significant levels of discomfort (4.0 on GCS) until they had sat for 240 continuous minutes. The ergonomically designed, and likely more comfortable, office chair participants used the office worker study might explain the difference in rates of increased discomfort observed in the two studies. Our findings are also consistent with those of Chester et al. who found 90 continuous minutes of sitting in an office chair in a laboratory was associated with reduced levels of physical comfort among a convenience sample 19 college-students [22]. The Chester study used a different measure of comfort and thus it is difficult to compare rates of impaired discomfort. 

Classroom sitting time was also associated with increased levels of sleepiness over time. Again, few studies have examined the relationship between uninterrupted sitting and alertness/sleepiness among college students. Still, our findings are consistent with those of Chester et.al. who found 90 continuous minutes of sitting in an office chair was associated with increased reports of tiredness among 19 college-students [22]. In a more recent study Mazzoli et al. explored the relation between classroom sitting time and attention over two class days among 149 first and second grade children (mean age 7.7 years). They found that objectively measured sitting time (activPAL™) was associated with more lapses in attention ((*β* = 0.12, *p* < 0.05) [23]. 

Understanding the relationship between classroom sitting time and measures of alertness and attentiveness is important considering the known inverse association between alertness and academic performance [24]. Students in the present study reported significant levels of sleepiness (3.0 on SSS) after just 15 min of class. However, this finding should be considered with caution as students entered the class reporting a sleepiness level of 2.5. Still, reports of sleepiness increased a full 1.0 units within 50 min of uninterrupted classroom sitting. It is also important to consider the quality and engagement of instruction students received during the lecture they were attending as these variables likely impacted student’s sleepiness but were not measured. Collectively, these findings support future interventions aimed at testing the effect of interrupting classroom sitting time as a means of maintaining student alertness during class. 

Pain experiencers entered class at a higher level of discomfort and reached the threshold of discomfort earlier than non-pain experiencers. This finding is important given 68.5% of participants reported experiencing pain in at least one area of the body over the past 12 months. These findings are consistent with Hupert et al. [25], who found 67% of college participants experienced pain or discomfort in the neck, shoulder, arms, wrist or fingers, indicating that many participants experience some form of pain. Implementing regular breaks from classroom sitting might be especially beneficial for those students who experience pain on a regular basis. This recommendation falls in line with current recommendations, which state sitting time should be broken up between every 5 min to 45 min [26].

Contrary to our hypothesis, sitting more during the day prior to class did not significantly impact either physical discomfort or feelings of alertness. It is possible that the activity required to get to the observed classroom immediately prior to class (e.g., walking a minimum of two blocks and climbing two flights of stairs) was sufficient to ‘reset’ feelings of discomfort and alertness such that students looked forward to sitting when they arrived to class. 

There are limitations to this study, which warrant caution when interpreting the findings. First, the small convenience sample of 54 students from a single classroom at a single university limits the overall generalizability of our findings. Further, the class was taught by a single instructor and it is possible that the instructor did not sufficiently engage the students, which could have impacted student reports of alertness. It is also possible that the timing of the class, which was taught in the late afternoon, also impacted reports of alertness and discomfort. In order to confirm our findings, our study needs to be replicated in other classes taught by other instructors at varying times of the day. Second, alertness and discomfort were measured with self-report measures which are subject to bias. Future studies that use objective measures of alertness (i.e., skin temperate [27] or Psychomotor Vigilance Test [28]) are warranted. Finally, we did not collect information regarding any participant’s habitual levels of physical activity or sports participation, which could be confounding variables that impact the entering levels of discomfort and/or pain of students. 

There were also several strengths of our study. First, this study was the first to explore the impact of uninterrupted classroom sitting on alertness and discomfort among college students. The study was also conducted in a real-world classroom setting which we considered to be a typical college classroom environment. Finally, the repeated cross-sectional design allowed us to explore patterns of discomfort and sleepiness over time and to determine the duration in which significant impairments in discomfort and sleepiness were reported.

The findings from this study should be used to inform future studies that examine the feasibility, acceptability, and efficacy of interrupting prolonged classroom sitting to prevent impairments in sleepiness and discomfort among college students. Future studies should seek to determine the optimal frequency (minutes in between interruptions), intensity (standing, walking, cycling), time (minutes of interruption) and type (stretching, traditional exercise, active learning exercises) of interruptions to be implemented in college courses. For example, our team found introducing sit-stand desks into college classrooms to be acceptable among college students [11]. In a follow-up intervention study, we found that introducing sit-stand desks into college classrooms reduced classroom sitting time by 9.3% [29]. However, introducing sit-stand desks is likely not a feasible solution for all classrooms. Previous studies conducted among elementary aged students have found breaking up long lectures with brief classroom activities (e.g., group discussion, one minute writing exercise) is effective at maintaining student engagement to foster learning [30]. Brief physical activity breaks have also been shown to improve on-task behaviors of elementary students [31,32]. Future studies that test similar instructor-led activity breaks in college classrooms are warranted, as these approaches may be more feasible, scalable, and cost-effective solutions for interrupting classroom sitting time. 

## 5. Conclusions

The present study demonstrated that uninterrupted classroom sitting time is associated with significant increases in reported levels of physical discomfort and sleepiness among college students. Significant levels of discomfort and sleepiness were reported after 75 and 15 min of uninterrupted sitting, respectively. These findings support future interventions that test whether interrupting classroom sitting time with light intensity activity is effective for preventing increased discomfort and/or sleepiness among college students.

## Figures and Tables

**Figure 1 ijerph-16-02498-f001:**
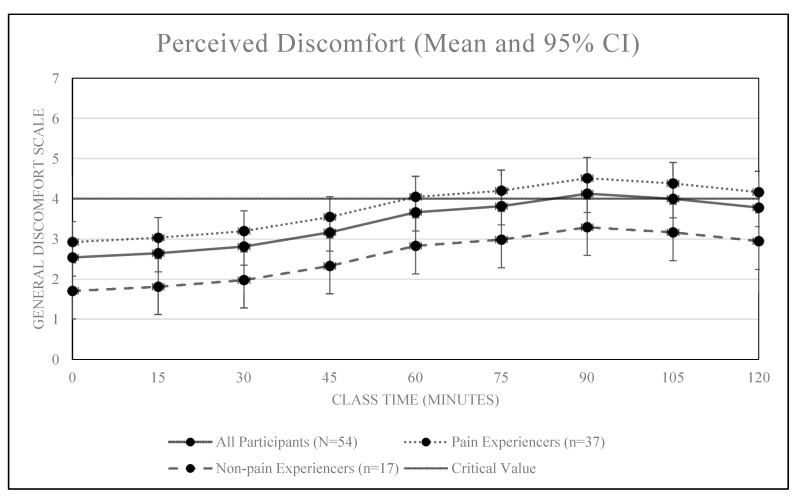
Mean discomfort scores over time by self-reported levels of pain.

**Figure 2 ijerph-16-02498-f002:**
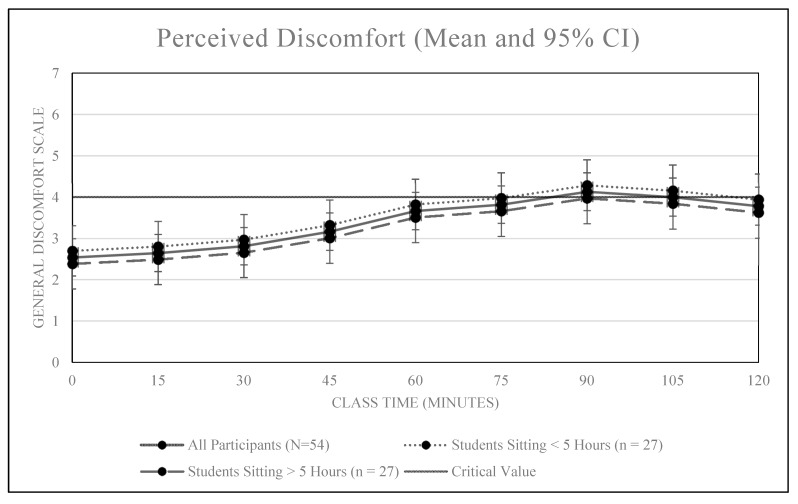
Mean discomfort scores over time by sitting time prior to class.

**Figure 3 ijerph-16-02498-f003:**
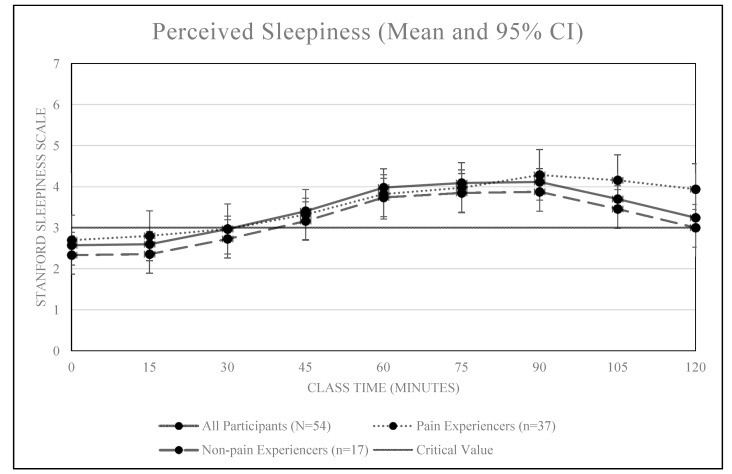
Mean sleepiness scores over time by self-reported pain.

**Figure 4 ijerph-16-02498-f004:**
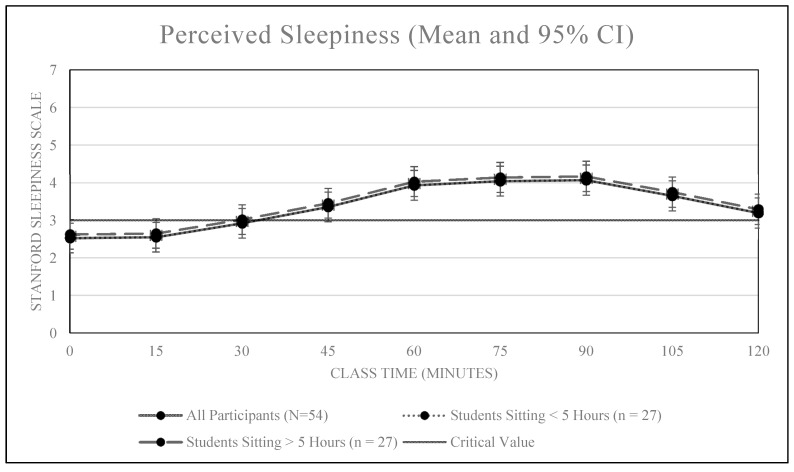
Mean sleepiness scores over time by sitting time prior to class.

**Table 1 ijerph-16-02498-t001:** Participant demographics and descriptive characteristics (*N* = 54).

Descriptive Variables	Mean (SD) %
Age (years)	20.3 (2.9)
Female %	67.3
Body Mass Index (kg/m^2^)	24.0 (4.0)
White (%)	78.2%
Average time awake prior to class (hours)	6.9 (1.8)
Average sitting time prior to class (hours)	5.1 (2.5)
Number of Pain Sites (0–7 range)	2.1 (1.8)
Percent students reporting pain in at least one area (%)	68.5
Baseline Alertness Score (1–8 range)	2.6 (1.1)
Overall Alertness Score (1–8 range)	3.4 (1.3)
Baseline Discomfort Score (0–10 range)	2.5 (1.6)
Overall Discomfort Score (0–10 range)	3.4 (1.8)

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
