# Peer review of "Uninterrupted Classroom Sitting is Associated with Increased Discomfort and Sleepiness Among College Students"

_ijerph, 2019, doi:10.3390/ijerph16142498_

Round 1

Reviewer 1 Report

The study is quite original and the results are useful to support changes in the way of programming classes in similar contexts.

The methods are clearly explained and presentation of the results respond to the objectives.

Although the researchers mention their limitations, I consider that the sample of 54 students from only one class is too small and insufficient to generalise results.

It would be interesting to repeat the study under similar conditions in other student facilities to increase the sample in a way that could allow generalising the results. Another limitation not mentioned relates to the practice(or not) of sports among young people, which could be a confounding factor for the study, given that the physical and mental condition of athletes differs from those who do not practice sports. In general, they are better fit, they probably present less discomfort and pain, and they may be able to maintain better attention with less sleepiness.

Author Response

Reviewer 1

The study is quite original and the results are useful to support changes in the way of programming classes in similar contexts. The methods are clearly explained and presentation of the results respond to the objectives. Although the researchers mention their limitations, I consider that the sample of 54 students from only one class is too small and insufficient to generalise results. It would be interesting to repeat the study under similar conditions in other student facilities to increase the sample in a way that could allow generalising the results.

We sincerely thank the reviewer for the positive comments on our study. We agree with the reviewer that this study is original and provides useful information that could impact how educators program their courses.  We also agree the small sample size limits the overall generalizability of the findings and we have noted this in our limitations section. We have added the following sentence to our limitations section on line 278: There are limitations to this study which warrant caution when interpreting the findings.

While we wish we could be more responsive to the reviewer’s request of repeating the study, we are simply unable to do this at this time due to limited resources. We hope that others will replicate our study to determine whether our findings can be confirmed and we have made this recommendation in our limitations section.

Additionally, while we are not able to generalize our findings, we do feel the sample size was sufficient to answer our primary question of interest (i.e., to determine the relations between prolonged classroom sitting and students’ perceived levels of physical comfort/discomfort and alertness/sleepiness). It is quite common for studies examining relations between continuous sitting and outcomes like discomfort and alertness to have sample sizes much smaller than our sample of 54.  For instance, our team recently demonstrated acute impairments in discomfort following four hours of continuous sitting among 15 middle-aged sedentary office workers in a laboratory setting (Benzo et al., JOEM, 2018). We reference this study on line 237.  Chester et al. demonstrated acute impairments in both discomfort and tiredness following 90 minutes of continuous sitting among 19 college-aged students (Chester et al., IJIE, 2001).  We make reference to this study on lines 242 and 248. These studies are commonly conducted in controlled laboratory settings which run the risk of introducing a Hawthorne effect.  For this reason, we took the unique approach of conducting our experiment in a natural classroom setting where students were being exposed to an actual lecture and sitting in their usual classroom environment. We see this as a strength of the study and feel this study adds to the literature in a significant way. We hope the reviewer agrees.

Another limitation not mentioned relates to the practice(or not) of sports among young people, which could be a confounding factor for the study, given that the physical and mental condition of athletes differs from those who do not practice sports. In general, they are better fit, they probably present less discomfort and pain, and they may be able to maintain better attention with less sleepiness.

We thank the reviewer for bringing this to our attention. We agree that student’s habitual levels of physical activity (including sports participation) could confound student’s chronic levels of physical discomfort and pain. Regrettably, we did not collect information related to physical activity or sports participation in this study and thus cannot reanalyze the findings to confirm this. We have added the following statement to our limitations section on line 287: Finally, we did not collect information regarding participant’s habitual levels of physical activity or sports participation which could be confounding variables that impact student’s entering levels of discomfort and/or pain.”

Reviewer 2 Report

The article has a suitable design for the objectives, the results are well presented and also the discussion, making clear the limitations of the study. Only some commets:

- About participants, all students had more than 18 years old? If not how managed the informed consent of student less than 18 years.

- At methodology section, statistical analysis, you showed some results (Table 1), I recommend move these results to the corresponding section (Results).

Author Response

Reviewer 2

The article has a suitable design for the objectives, the results are well presented and also the discussion, making clear the limitations of the study. Only some comments:

- About participants, all students had more than 18 years old? If not how managed the informed consent of student less than 18 years.

All students enrolled in this class were older than 18 years of age. We have added the following sentence to the Participants section on line 101: “All participants were at least 18 years of age.

- At methodology section, statistical analysis, you showed some results (Table 1), I recommend move these results to the corresponding section (Results).

Thank you for this suggestion. We have moved Table 1 to the results section as requested. 

Round 2

Reviewer 1 Report

The authors have given a satisfactory response to the comments, including paragraphs that support and explain the limitations of the study